# Empowering Young Women: A Qualitative Co-Design Study of a Social Media Health Promotion Programme

**DOI:** 10.3390/nu16060780

**Published:** 2024-03-09

**Authors:** Jessica A. Malloy, Joya A. Kemper, Stephanie R. Partridge, Rajshri Roy

**Affiliations:** 1Discipline of Nutrition and Dietetics, School of Medical Sciences, Faculty of Medical and Health Sciences, University of Auckland, 85 Park Road Grafton, Auckland 1011, New Zealand; 2Management, Marketing, and Tourism, University of Canterbury, Christchurch 8041, New Zealand; 3Charles Perkins Centre, The University of Sydney, Camperdown, NSW 2050, Australia; 4Discipline of Nutrition and Dietetics, Susan Wakil School of Nursing and Midwifery, Faculty of Medicine and Health, The University of Sydney, Camperdown, NSW 2050, Australia

**Keywords:** digital health, social media, young women, nutrition, behaviour change

## Abstract

Social media platforms may be promising intervention tools to address the nutrition literacy and associated health behaviours of young women. We aimed to co-design a lifestyle intervention on social media targeting eating, physical activity, and social wellbeing that is evidence-based, acceptable, and engaging for young women aged 18–24 years. The study used a participatory design framework and previously published iterative mixed methods approach to intervention development. Matrices for workshop objectives were constructed using expert discussions and insights were sought from young women in participatory workshops. A 10-step qualitative data analysis process resulted in relevant themes, which guided intervention development. The resulting intervention, the Daily Health Coach, uses multiple features of Instagram to disseminate health information. Co-created nutrition content considers themes such as holism, food relationships, and food neutrality and acknowledges commonly experienced barriers associated with social media use such as nutrition confusion, body image concerns, and harmful comparison. This study may guide other researchers or health professionals seeking to engage young women in the co-design of women’s health promotion or intervention content on social media.

## 1. Introduction

Social media platforms attain a significant share of youth attention and are therefore promising platforms to engage with young people, advocate for change, and disseminate health information [1]. Researchers and health educators have begun utilising social media to share health information with youth in recognisable and accessible formats [2]. Social media is a space where creativity and collaboration with other specialities including digital marketing, filmmaking, and web design are important to success [3]. Recent health promotion campaigns or interventions amongst young adults that have been effective and include the amyotrophic lateral sclerosis (ALS) ice bucket challenge, an Instagram intervention for young people to moderate portion sizes, and a novel intervention utilising “Webisodes” or entertaining and educational sexual health episodes uploaded to YouTube and Facebook [2,4,5]. 

The target demographic for this research is women aged 18–24 years in Aotearoa, New Zealand. To the best of our knowledge, no studies pertaining to social media nutrition interventions have been conducted specifically for this population group. For New Zealand adults, improvements in nutrition, physical activity engagement, and sleep are needed [6,7]. Healthy habits established during young adulthood can be consequential to future wellbeing [8]. This highlights the need to intervene and educate young women in Aotearoa (New Zealand) to increase engagement in healthy behaviours. 

Social media is often used as a health information source, which may lead to exacerbation of dietary quality concerns, as well as misleading perceptions regarding health and nutrition. The quality of nutrition information on websites and social media was evaluated in 2023 [9]. The comprehensive content analysis demonstrated the quality of information was “*low and often inaccurate*”, highlighting the risk of encountering misinformation. Nutrition mis- or disinformation and prolonged social media use have a number of established consequences for young women including body dissatisfaction, disordered eating behaviours, and a distorted sense of self [10]. These findings are associated with a recent shift in how young people view social media. Youth users of apps such as Instagram and TikTok report limiting their time on social platforms and acknowledge their consequences on mental wellbeing [1]. These aspects of health, nutrition, and social media were explored with young women to determine common sentiments and direct tool development. 

A number of systematic reviews have been conducted to assess social media as a tool for the promotion of healthy behaviours [11,12,13,14,15]. Regarding nutrition interventions specifically, a review conducted in 2018 found that social media is a promising tool for young people. However, the evidence base was limited and the majority of included studies did not evaluate efficacy [13]. A more recent review concluded that social media interventions had a positive impact on physical activity and diet-related behaviours [15]. Although feasible for engaging with youth, it is important to grow the evidence base and outline how best to utilise social apps as behaviour change tools, as well as evaluate their effectiveness. 

Co-design was selected as the principal methodology underlining intervention development due to the incorporation of user perspective and its adaptability. The dynamic nature of co-design allows for ongoing iteration according to user evaluation and feedback [16]. The inclusion of user perspectives and voices throughout the creation process can increase the likelihood of developing an efficacious tool for young people [3,17]. Co-design is a methodology that plays with power dynamics, placing leadership in the hands of those who will be impacted by decision making and outcomes [18,19]. Furthermore, user-generated content (UGC) is an important consideration for health promotion utilising social media [20]. Recent health campaigns have attested to the use of UCG as a success factor in their interventions [21,22]. 

## 2. Materials and Methods

This research sought to understand how to utilise influencer communication techniques to disseminate nutrition information to young women via popular social apps. The overarching objective was to co-design a health promotion program for young women in Aotearoa using social media. This aim was translated into the following research questions:Which social media components might work best?How should the program be delivered?What nutrition behaviours should the program aim to change?Who should deliver the program?

The final intervention is a 12-week healthy lifestyle education program for young women aged 18–24 years run on the social media platform Instagram. The content covers a range of topics including nutrition facts and information, body image and disordered eating awareness, social media use, food relationships, recipe sharing, and healthy behaviours. Information is disseminated in the form of posts, stories, reels, and direct messages. 

There is limited guidance for reporting the use of inclusive methodologies. This research follows the participatory process reporting of Cheng et al., as well as the Think NPC Toolkit and recommendations by the Young and Well Research Co-operative Centre and the Health Cascade Network [23,24,25,26]. The COREQ (consolidated criteria for reporting qualitative research) checklist has been used to guide the report of methods and findings [27]. A protocol has been published for the co-design, co-creation, and feasibility testing of the DHC, which provides a comprehensive overview of participatory methods [28].

### 2.1. Participant Recruitment 

A convenience sampling method was used to recruit participants via personal and organizational social media sites. Digital recruitment posters were shared by the research team, as well as the Faculty of Medical and Health Sciences at the University of Auckland, across LinkedIn, Facebook, Instagram, and Twitter. Recruitment ads (including physical posters placed around the university campus) directed prospective participants to the study website and screening survey to determine eligibility. 

Enrolment took place via email. Once enrolled, participants were asked to pass on research team contact details to connections within the target demographic, utilising the snowball sampling method to identify additional participants. Young women were recognised for their time and participation with NZD$50 supermarket vouchers for each workshop attended. The research was approved by the Human Participants Ethics Committee at the University of Auckland on 9 June 2022 for three years (UAHPEC24366). 

### 2.2. Participatory Design

Co-design guidelines published by the NPC and the Young and Well Research Co-Operative Centre were used to develop the participatory framework for the development of the Daily Health Coach intervention. The final framework included five key participatory phases (Table 1). 

The student researcher on the project (JM), who is a New Zealand registered dietitian (NZRD), hosted co-design workshops, focus groups, and interviews and was responsible for knowledge translation between phases as well as contributions to co-creation and content finalisation. All participatory activities were performed with oversight from the primary investigator (RR, PhD), who had prior experience in facilitating focus groups and workshops. 

### 2.3. Knowledge Translation

Participatory phases one and two were hosted on the conferencing software Zoom (version 5.16.10) [29]. Interviews, focus groups, and workshops were audio recorded and transcribed using Zoom and Otter AI. Transcripts were cleaned by the primary researcher. 

#### Data Analysis

Braun and Clarke’s and Kreuger and Casey’s strategies were utilised to guide data analysis [30,31]. This consisted of a ten-step process of knowledge translation. Two individuals coded the data separately; one for the purpose of tool development and one for research into the impact of social media on young women in Aotearoa. Findings were compared and discussed. For the purpose of tool development, the student researcher (JM) coded all data, meeting with the research supervisor (RR) frequently to discuss thematic synthesis. Initial deductive codes (*n* = 20) related to research questions and included cultural relevancy, gaps in nutrition research and education, and perceived or experienced barriers to maintaining healthy behaviours. Inductive codes were generated whilst deductive coding took place (*n* = 90). These included enablers to social media use, overcoming barriers, and health information-seeking behaviours. 

Coding and analysis were performed using NVivo Qualitative Analysis Software (version 12) [32]. Once inductive and deductive coding was complete and discussed with the research supervisor, similar codes were combined to generate high-level codes (*n* = 11). These high-level codes were used to generate themes. The final themes were guided by the research question and included body image, co-creation, engagement, simple and sustainable changes, holism, and prioritising self, amongst others. Co-design participants were invited to attend a presentation of findings; however, no responses were received. 

## 3. Results

### 3.1. Participant Characteristics 

Due to university convenience sampling methods and the age range of the targeted demographic, the majority of participants (63%) were university students (Table 2). All women attended at least two workshops. In total, 16 of the 19 young women attended all workshops, resulting in a full attendance rate of 84.2%. Of those who missed a workshop (*n* = 3), one forgot to attend, one reported feeling unwell, and one did not provide a reason. 

Perceptions of body image and experiences with body shaming may differ based on an individual’s body shape [33]. Information on body type and weight was not collected from co-design participants due to a core pillar of this research being body neutrality with non-weight-centric goals. We therefore cannot comment on distinct perceptions and experiences of body image and shaming associated with different body types.

Co-design participants were asked to complete the revised 18-question Three-Factor Eating Questionnaire (TFEQ-R18) as part of the recruitment screening process. The TFEQ-R18 measures cognitive restraint and uncontrolled and emotional eating behaviours [34]. This was administered to ensure the research team had awareness of participants who may find conversations particularly triggering. The majority of participants (18/19) did not demonstrate concerning disordered eating behaviours (score of >75 for each measure). 

### 3.2. Phase 1: Expert Focus Groups and Interviews

A visual representation of high-level coding and classification demonstrates key insights obtained from early expert discussions has been included as a Appendix A. Interviews lasted from 30 to 45 min; focus groups ran for 1–1.25 h. Interview questions for marketing professionals or academics were structured to ascertain a more comprehensive understanding of social media algorithms and digital marketing practices. As well as audio-recording, field notes were taken by the student researcher during interviews and focus groups where necessary. Key insights from these conversations included the importance of engagement to ensure social algorithms’ push content to consumer newsfeeds, advised metrics of success, and discussion of targeted ad use. 

Conversations revealed that the small size of the pilot-testing intervention (*n* = 25 participants or ‘followers’ at any given time) presents a unique challenge. Further to this, the ‘closed’ nature of the intervention, meaning a ‘private’ account, creates a further barrier to disseminating content. A private account is required to be used in order to avoid contamination of research samples when pilot-testing the program on social media. Both of these factors made the use of targeted ads redundant. The suggested strategy to overcome the discussed barriers involves transparency with young consumers to pseudo-engage with content seen in the first two weeks of the programme. This helps to ‘teach’ the social algorithm to continue pushing content to individual newsfeeds. 

Due to the use of a private account, scheduling tools are unable to be used. This places a large burden on those ‘running’ the account to draft and post all content daily at pre-determined times. It also means that data insights obtained from the Facebook Business Suite were not an option for this research. Insights on user-generated content were obtained from all marketing professionals. The common sentiment of UGC was that it is beneficial to campaigns and its use was advised where possible. 

Distinctions between traditional vs. social media marketing was outlined, highlighting the utility of two-way dialogue. It was suggested that this capability of social media be tapped into to create a sense of community and a safe space for learning and sharing.

“The company spends a lot of time trying to create a community—so having a lot of back and forth, which means we often put question boxes up saying “What do you want to see next? What colours? What styles?”—a digital marketer.

Experts were asked which social media metrics should be monitored in the determination of intervention success. This resulted in an outlining of engagement vs. vanity metrics. It was suggested that outcome measures should focus on engagement (comments, shares, and replies), rather than likes and views. 

Focus group conversations with New Zealand-registered dietitians and nutrition researchers resulted in a deeper understanding of nutrition-related habits, behaviours, and pain points of young people from the practitioner’s perspective. Using a moderation guide, gaps in nutrition intervention and health promotion for this age group were discussed. Important insights from nutrition professionals included the pertinence of “*for youth, by youth*” regarding cultural relevancy and relatability of online nutrition information, the utility and disadvantages of social media for the health of young people, and aspects of nutrition to explore in proceeding co-design workshops. Suggestions for questions to bring to young women were valuable to the structuring of workshops and activities.

“You need to know what features of the social media applications they use most frequently.”—a NZ-registered dietitian;

“Understanding what they’re using social media for. You know, whether they are using social media for health information.”—a NZ-registered dietitian

“Seeing what sort of health content they are engaging with. And asking them, is there something else that would motivate you to engage with it? And what are any barriers to using social media? I would investigate what are the barriers to using social media for health-related information. Because it might be more of an access issue.”—a NZ-registered dietitian;

“What are the problems? What do they want to know more about? What are the pain points around their well-being? What do they feel they need to work on to achieve better health, and how important is that to them?”—a NZ-registered dietitian.

Nutrition professionals also highlighted the utility of behaviour change models, such as the COM-B model, to underpin developed content and engagement strategies [35]. This would involve consideration of consumer capability in the physical, psychological, and social realms. No repeat interviews or focus groups were carried out. 

### 3.3. Phase 2: End-User Workshops

Workshop activities and objectives are outlined in Table 3. High-level thematic analysis codes, themes, and research questions are represented visually in Appendix A. Key insights have been categorised by workshop session. 

#### 3.3.1. Session One

The beginning of the first workshop was dedicated to ice-breaking and outlined the personal and professional backgrounds and goals of each researcher. Decision-relevant information from the first workshop concerns the meaning of health and wellbeing to young women and perceived barriers to achieving or maintaining healthy behaviours. Reflection sessions revealed a multi-faceted view of health, including community support, sleep, mental wellbeing, physical activity, balanced eating, gratitude/appreciation, Te Whare Tapa Wha Māori model of health, strong connection with self and with others, safe housing, vitality, faith, and accessible healthcare [36]. 

“Doing things that make you feel good, not focusing on what makes others feel good, or what you think others would like.”—young adult 4;

“Connecting with yourself to connect with others. I think that’s really important.”—young adult 5;

“The words that I would use to describe health would be variety and unlimited.”—young adult 2.

The commonly identified barriers to achieving health goals included time, motivation and accountability, finances, support, knowledge, self-care and self-worth (body image), food relationships, mental illness, comparison and social media, stress, and confusion regarding nutrition information. When viewing different nutrition content on social media, participants acknowledged the perpetuation of beauty standards and the importance of relatability (including the incorporation of local food products vs. international products not available in New Zealand), gatekeeping in nutrition, authenticity, and credibility. 

“I think an issue at the moment is that there’s too much emphasis on perfection when it comes to health.”—young woman 3;

“I just find understanding nutrition like really difficult, like I just don’t get it”—young woman 8;

“Health is not all one size fits all, and it’s really, really difficult to navigate life constantly comparing yourself and your own health to what other people eat in a day or how much other people exercise, or how other people view themselves in their bodies.”—young woman 12.

#### 3.3.2. Session Two

The second session gathered important insights on social media as a platform for behaviour change. Overall, young co-designers believed that social media could help young people to change their health behaviours. Young women in this study suggested that social media could be used to share “visual guides” for simple changes and healthy habits using exemplar local food products, underlining messages with a focus on addition rather than restriction (sustainable changes). Other suggestions included sharing easy ways to get in more exercise each day (for example, taking the stairs), simple and affordable recipes (including prices, ingredient lists, and instructions on the post), sharing information on the seasonality of foods, sharing other credible information sources to follow on social media and elsewhere on the internet, addressing mixed messages in nutrition, acknowledging the pitfalls of social media, direct messaging for accountability, authentic and relatable information (via infographics), acknowledgement of budgets and financial constraints, and inspiring reels.

“It is quite refreshing to see videos with quite normal ingredients. Quite often, when I get these videos on my social media, it looks amazing but it’s really expensive ingredients and takes a lot of time.”—young woman 11;

“I feel like with health… I find that there is a lot of different information, even if it’s with a credible source?”—young woman 13.

Social media is used by the cohort for a myriad of reasons (Figure 1). Commonalities in the use of social media included entertainment purposes, social connectivity (including to cultural, religious, and other community groups), and seeking information and inspiration. 

#### 3.3.3. Session Three

The final workshop gained insight into participants’ perceptions of different nutrition influencers across Instagram and TikTok. Key pillars of ‘influence’ included a healthy mindset and positive undertone of messaging (balanced, inclusive, and non-diet/non-weight centric). 

Finally, participants were placed into breakout rooms and asked to brainstorm what they believe the intervention should look like, including the name, platforms of use, type of content posted, and achieving inclusivity. Overall, participants agreed the name of the intervention should remain. However, they concluded that due to the name being the ‘Daily Health Coach’, content should be posted on a daily basis. 

#### 3.3.4. Social Media Features and Strategy

A number of social apps were suggested to be used as intervention platform(s). These included but were not limited to Facebook (including Messenger), Snapchat, TikTok, Pinterest, YouTube (including shorts), and WhatsApp (group chat functions). 

During thematic analysis, it was apparent that the most commonly suggested platform was Instagram, followed by TikTok. The ability to run a multi-platform intervention for two cohorts was discussed by researchers at length. It was decided that TikTok videos would be created and posted as reels to Instagram. This way, TikTok-style videos can be evaluated without requiring participants to check content on multiple apps. Furthermore, workshop participants were asked to vote on which app to use and provided screen time information. Instagram was selected by most as the app of use, as well as the most used app according to screentime metrics. This is aligned with survey findings of 350,000 social media users outside of China in 2023 [1]. 

Young women in this study suggested that pilot-testing participants have the ability to select how often they would like to be contacted by the research team. This was due to unwanted messages or notifications being perceived as “scammy” or unprofessional. A combination of posts, reels, and stories was advised to cover bases and ensure every type of Instagram consumer was being provided with their preferred style of content. A parallel SMS or Slack channel as a separate more formal forum for instigating conversation about goals and progress was discussed. This aligns with expert recommendations that discussion is an important aspect of retaining health information.

“We weren’t too sure about the contacting participants… not going to lie… but we thought maybe, I don’t know, we thought don’t DM people because they will think it’s a scam, so you might not get answers”—young woman 1.

#### 3.3.5. Information Delivery 

Two themes arose when ascertaining the best person(s) to deliver the platform: co-creation and credibility. Relatability, authenticity, and relevancy were discussed as important factors for effective health-related social media content. The idea of a youth advisory group or collective of content creators was discussed to incorporate multiple perspectives in content creation. 

Further to this, in a space of rampant misinformation, credibility was repeatedly highlighted as an important value in content creation and information sharing. This involves ensuring creators of content are qualified to discuss and deliver health information. To address this, student dietitians were invited to co-create content for the DHC. All content created was recorded alongside evidence cited. Cited evidence was uploaded to the DHC website, accessible to those participating in the program. 

#### 3.3.6. Other Insights

Due to changing sentiment regarding social media use, it was important to ascertain user perceptions about their screen time. Young women were asked about their barriers and enablers for using social media. They were also polled on their use of app limits to reduce screentime. 

Some identified barriers to using social media included perpetuated beauty standards and harmful comparison, misleading or inaccurate nutrition and/or health messaging, and information overload. 

“That is a real barrier—understanding what the standards are that are being pushed on us on social media vs. who am I?”—young adult 8;

“I feel like at some point it’s kind of like trying to confirm if I was correct? Because, you know, I might have some values, I might have some knowledge about this particular topic, but it’s just me and, you know, it will be really nice if I can look out to a platform that is so easily accessible, and there are people up there like, regardless of them being right or wrong or not, but you know we’re all on the same topic.”—young woman 2;

“I found that a massive barrier is that I don’t invest as much ‘in person’ time with my friends as much as with social media.”—young woman 8.

Enablers of social media use were accessibility (passive consumption; on the same device as other tasks such as study or work; and social media use is an established daily habit for most young people), convenience (for example, when messaging and graphics portray health information in simple and entertaining ways, simple recipe videos with instructions—aspects of health dissemination that make things simpler), and validation of ideals or beliefs. The addictive nature of social media was identified as both a barrier and an enabler by one participant. Most co-designers did not have time limits set on their social apps (81.8%).

### 3.4. Phase 3: Content Co-Creation

Each student dietitian was responsible for creating a week’s worth of content. This involved the creation of seven Instagram posts, seven Instagram stories, and seven Instagram reels or TikToks. Guidelines for the development of content are aligned with recommendations by Denniss et al. using the PHRISM framework [37]. All content was created using the graphic design software Canva (www.canva.com; accessed on 30 August 2023). Posts follow the same neutral template. For ease, each day of the week was separated into a theme (Table 4). Sample carousel title posts from the ‘Real Talk Friday’ and ‘Things You Should Know’ content themes have been provided (Figure 2 and Figure 3). The content of each daily post is discussed by the post creator in the form of an Instagram story, which the research team refer to as ‘story explainers’ (Figure 4). The Instagram reels which were created are associated with the content topic of the day, however, it was suggested that student dietitians get creative and present the information in fun and engaging ways (Figure 5). In total, 252 items of content were created: 84 Instagram stories, 84 Instagram reels, and 84 Instagram posts. 

All content considers a holistic view of health and takes a non-diet food-neutrality approach. As suggested by co-designers, suggestions are simple, realistic, and sustainable. Key messages of the content reinforce that there is no ‘one size fits all’ with the diet, encouraging exploration and addition of nutrient-dense foods rather than a focus on restriction. Furthermore, it is suggested that sustainable change should come from a place of self-compassion. For those interested in learning more, a list of dietitians and nutritionists on social media who share evidence-based mindful dietary advice will be shared with followers (referred to as ‘healthy handles’), as well as podcasts, books, and body-positive influencers.

#### The Daily Health Coach Website

As aforementioned, the idea of a separate space for sharing resources with followers was discussed with the cohort. The website used in the recruitment of young women was transformed into a hub of information for future followers of the Daily Health Coach (Figure 6). This website was created using Wix and has all DHC recipes, healthy handles (evidence-based nutrition content creators), a couple of longer-form videos, and the evidence used in the creation of each post [38]. It was important to share the evidence used in content creation for followers to view at their leisure, as well as for reassurance of evidence-based information sharing. All webpages are password-protected, only to be accessed by those within the DHC programme. 

### 3.5. End-User Feedback

Feedback received after the final workshop was positive. One nutrition graduate from the co-design workshops created a post for the intervention and provided feedback on the created content. This feedback was delivered to a student dietitian, who outlined the incorporation of feedback into the final intervention. Suggestions included rephrasing “healthy fats” to “nutritious fats” and removing title slides from Instagram stories, among others. This feedback was incorporated prior to the final rollout. 

“Thanks for including me in your study, I really enjoyed being a part of it”.

“It was a really great experience to share my ideas around health and food with a small group. I definitely learnt a lot about other’s perceptions and how I could incorporate little habits to make my mealtimes healthier”.

“Thanks for the opportunity! I really enjoyed getting involved”.

## 4. Discussion

### 4.1. Principal Findings 

The results affirm that social media is a desirable platform for behaviour change in young women and offers important guidance in the development of social media health interventions targeted at youth who identify as female. Overall, young women co-designing the DHC believe social media can be used to positively influence health behaviours. This is encouraging, as the promise of social media is appreciated by young consumers as well as those working in the health promotion space. 

Cohort views of wellbeing went beyond healthy eating to acknowledge a myriad of influences on health including mental and social wellness, community support, spirituality, and self-worth. These “non-food” holistic factors provide important insight into external considerations when discussing nutrition and creating evidence-based nutrition information for dissemination to young women. 

The pitfalls of social media including rampant harmful body ideals and conflicting nutrition information were organically acknowledged and have been considered for the resulting intervention alongside food relationships and general dietary confusion. As researchers, we feel compelled to acknowledge the negative aspect of the tool we have decided to use as our behaviour change platform and find ourselves attempting a balancing act of sharing nutrition information in spaces frequented by the target group, whilst acknowledging the consequences of extended use of the app. To do so, we will be suggesting non-tech self-care activities, providing evidence-based handles across the platform for participants to follow, offering advice on identifying nutrition misinformation, and encouraging critical thought when it comes to dietary information on social media.

### 4.2. Comparison to Similar Research

A number of health interventions using social media have been co-designed with young people and have utilised platforms such as Facebook Messenger and Instagram [39,40,41,42]. Other promotion strategies within academic research have involved collaboration with film makers to develop short educational films [43,44]. For example, the short films for the “What’s Up with Everyone” project were disseminated via YouTube [43]. Recent evidence indicates digital health interventions will not see success with little or no engagement [40,45]. Of note for the Daily Health Coach intervention, an ‘adjunctive strategy’ for engagement was developed with young people aged 17–24 years [40]. 

The strategy developed in 2023 by Gan and colleagues uses an Instagram page to encourage young people to engage with the mental wellness app ‘LifeBuoy’ [40]. Strategy co-design findings align significantly with those of DHC development, including positive perceptions of social media use for behaviour change, Instagram as the intervention platform of choice, and utilising the app to create a sense of community. Content themes were also similar, with Lifebuoy co-designers advocating for information that is “*practical and easy to use in everyday life, uplifting and motivational, and emotionally validating*” [43]. Important considerations for the DHC include the timing of direct messages (LifeBuoy uses suggested either weekends or outside of working hours) and a “*calming*” aesthetic [43]. 

### 4.3. Relevance and Contribution of Research

This study adds to the growing evidence base for youth social media health interventions. The research offers important insight into the perspectives of young women when it comes to social platforms, health and nutrition, and the utilisation of social apps for behaviour change. To the best of our knowledge, the Daily Health Coach is the first nutrition-focused social media health promotion program created specifically for young women in Aotearoa New Zealand. The need for evidence-based information when it comes to nutrition is evidenced by harmful associations of prolonged social media use including body image disturbances and disordered eating behaviours [10]. It is important to note that the Daily Health Coach content was created by individuals who obtain, at minimum, a bachelor’s degree in nutrition. However, the literature gap of efficacy was not addressed by this study. We aim to pilot test the Daily Health Coach in a feasibility trial with young women from across New Zealand (*N* = 50), which will add to our collective knowledge of the effect of social media interventions on young people. 

### 4.4. Strengths and Limitations

This study is among the first of its kind in the field of digital health promotion. Our research offers valuable insight into the desires of young women with regard to healthy behaviour change via social media. The participatory process guided the translation of common communication techniques utilised by influencers across social platforms. These techniques are recognisable and familiar to young women and may help to encourage positive dietary change. 

Co-design offers individuals the ability to contribute to local research and affect change [46]. Young people may find benefit from co-design participation such as gained knowledge and confidence, particularly concerning research outputs that impact them and by which they are targeted [47,48]. The dynamic nature of co-design allows for ongoing iteration, which may see final outputs that better align with consumer desires and needs [46,49]. The next stage of research, whereby the DHC is trialled for feasibility, will allow for further changes based on end-user evaluation. 

There are limitations in the scope of transferability of our findings. When applying the research findings, it is important to bear in mind the lack of representation of the co-design group. Application of the shared insights is likely inappropriate in distinct contexts. The cohort for the development of the DHC is not culturally or socio-demographically representative of the population of young women in Aotearoa, New Zealand. The majority of co-designers who attended workshops live in Auckland hold at least a bachelor’s degree, with many receiving an education in nutrition (Table 1). 

Due to the target demographic of young women, the findings are not applicable to other genders or non-gender-conforming individuals. Participation by young women who identify as Māori, Pasifika, and MLAA (Middle Eastern, Latin American, and African), as well as those with distinct educational and environmental backgrounds (e.g., young women living in rural locations), is needed for the DHC intervention to be culturally and socio-demographically representative. 

In Aotearoa, significant health disparities exist between Pākehā (European) and Māori and Pasifika populations [50]. Colonisation and systemic racism are key drivers of inequitable access to social determinants of health including education, housing, and healthcare [51]. To create a truly culturally relevant and effective tool and practice Te Tiriti o Waitangi-based health promotion, whanaungatanga (relationship building through shared experiences) with Māori and talanoa with Pasifika co-designers in every developmental stage of interventions such as the Daily Health Coach is fundamental to understanding and incorporating Te Ao Māori and Pasifika experiences, perspectives, and cultural knowledge in research outputs [52]. We advise adaptation of the associated protocol for this research for use by distinct population groups nationally and internationally, rather than direct application of the co-design findings [28]. 

The impact of personal bias, both unconscious and conscious, is important to note in conducted qualitative research where collated transcripts are coded and surmised. There is always a likelihood of negative outcomes such as bias in co-design despite best efforts to mitigate its effect [53]. For example, the outline of the DHC intervention was drafted prior to co-designing workshops and focus groups to obtain ethics approval. Despite being open to adaptation of all aspects of the DHC during co-design phases, it has been suggested that co-design should go a step beyond this, whereby the research objectives and questions are established from the outset by co-designers where possible, rather than research teams. 

Finally, a systematic evaluation of the co-design process on participants was not performed. Obtaining an understanding of the effect of co-design on young designers would involve ascertaining, for example, whether engagement in workshop discussions resulted in feelings of empowerment or changes to health behaviours [47]. These findings would add to the evidence base on the merit of co-design beyond intervention output.

## 5. Conclusions

The co-design methodology was utilised to develop a social media health promotion program *for* young women, *by* young women. The resulting intervention utilises the social networking platform Instagram to disseminate health information in familiar formats including posts, reels, stories, and direct messages. Content covers a range of topics desired by co-designers including simple and affordable recipes, actionable nutrition information, and advocation of sustainable, non-diet approaches to behaviour change.

Regarding health influencers on Instagram, nutrition professionals (dietitians, board-registered nutritionists, medical doctors, and academic researchers) remain significantly outnumbered. The DHC intervention offers evidence-based, credible, and reliable information identified as essential to young women. This comes at a time where misinformation awareness on social media platforms can make it difficult for consumers to trust information encountered online. The DHC will act as both a source of reliable nutrition information as well as a platform for the advocation of other nutrition professionals on social media. 

Finally, DHC content acknowledges and will attempt to instigate conversation regarding identified nutrition barriers and pain points commonly experienced by young women, including; body image, self-compassion, food relationships, and harmful comparison. Despite increasing conversation and awareness of these contributors to nutrition habits, the identified factors continue to play a concerning role in women’s wellbeing and should be acknowledged when discussing and encouraging healthy behaviours. 

### Future Applications 

There remains a pressing need to determine the benefits of co-designed health interventions beyond accessibility. The DHC intervention is presently being trialled for feasibility and health behaviour change, with results expected in the second half of 2024. The programme will be adapted according to pilot results with the intent to roll out the intervention to a larger campaign for young women across Aotearoa, New Zealand. Future nutrition interventions for young women may wish to consider other contributors to wellbeing and eating behaviours, such as body image, food relationships, and confusion when it comes to nutrition information and scepticism regarding credibility and trust. Participatory methods should be employed when developing digital health interventions for young people to increase the likelihood that tools are feasible, accessible, and usable. Finally, those who develop youth digital health interventions via social media should evaluate the health impact of their tools on their target demographic. 

## Figures and Tables

**Figure 1 nutrients-16-00780-f001:**
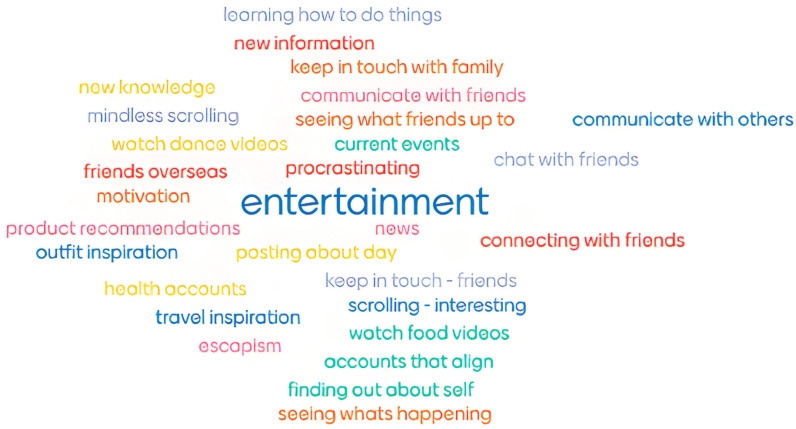
Thursday workshop social media use of a word cloud.

**Figure 2 nutrients-16-00780-f002:**
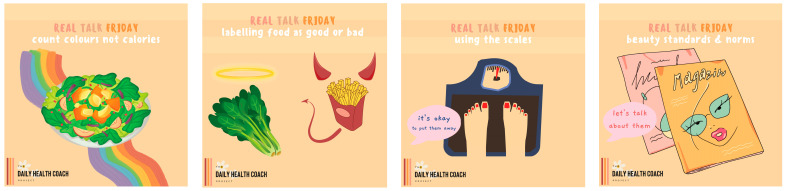
Carousel title posts for real talk content [Instagram post].

**Figure 3 nutrients-16-00780-f003:**
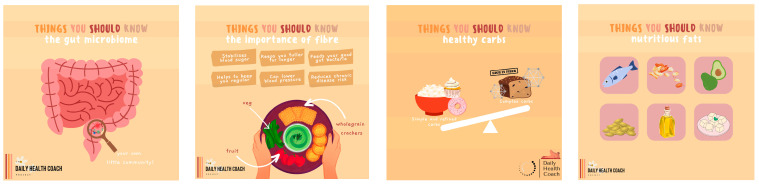
Carousel title posts for nutrition information content [Instagram post].

**Figure 4 nutrients-16-00780-f004:**
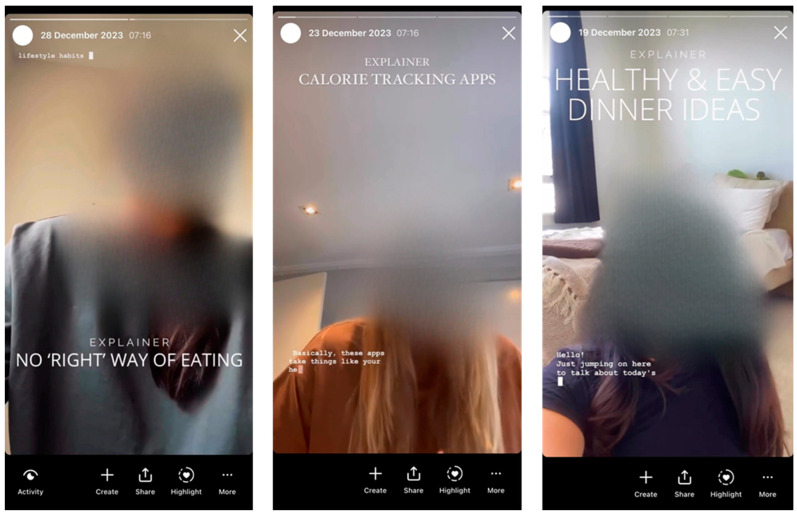
Story explainers [Instagram stories feature].

**Figure 5 nutrients-16-00780-f005:**
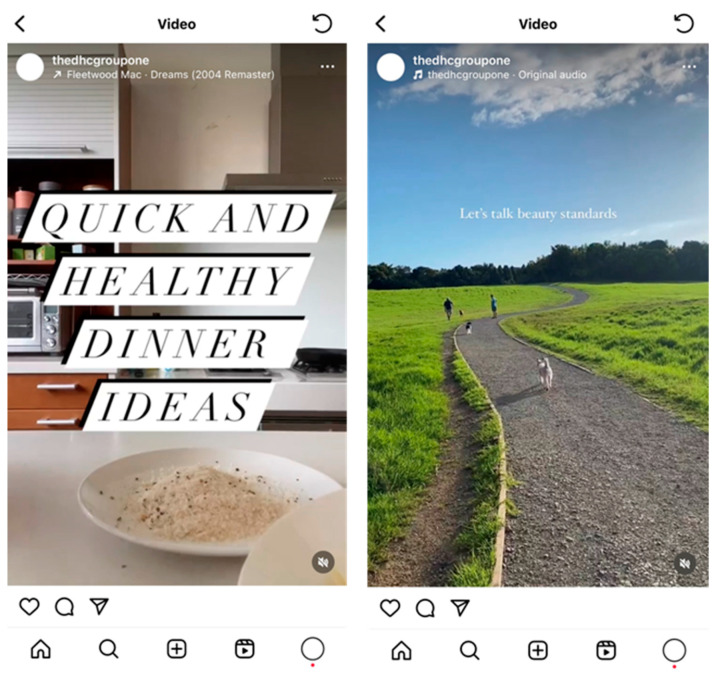
Associated reels [Instagram short video feature].

**Figure 6 nutrients-16-00780-f006:**
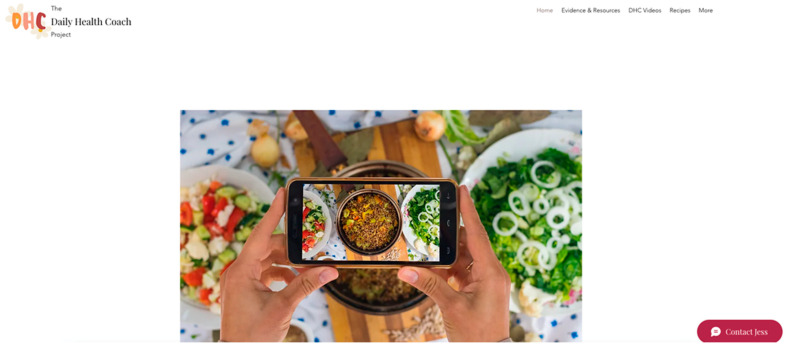
The Daily Health Coach resources website.

**Table 1 nutrients-16-00780-t001:** Key participatory phases of the Daily Health Coach development framework.

Phase	Participatory Activity	Participatory Stakeholders	# of Sessions
1	Expert Focus Groups and Interviews	Six dietitians, one nutrition researcher, two marketing academics, and three digital marketers.	Two focus groups and five one-on-one interviews.
2	End-User Co-design Workshops	Nineteen young women aged 18–24 years.	Three workshops, offered in three slots over three weeks.
3	Content Co-Creation	Ten student dietitians from the University of Auckland and one nutrition graduate from the co-design workshops.	
4	Expert and End-User Evaluation	One Senior Lecturer in Marketing, one Senior Research Fellow and Dietitian, and one participant from the co-design workshops.	
5	Content Finalisation	One student dietitian.	

**Table 2 nutrients-16-00780-t002:** Co-design workshop participant characteristics (*N* = 19).

Median Age (yrs)	Location	Ethnicity	Occupation
21.4 ^a^	Auckland, NZ (*n* = 17)	NZ European (*n* = 8)	University Student (*n* = 12)
	Waikato, NZ (*n* = 1)	Indian (*n* = 3)	Financial Sector (*n* = 3)
		Chinese (*n* = 3)	Food Industry (*n* = 2)
		Fijian Indian (*n* = 2)	Health Research (*n* = 1)
		Hong Kong Chinese (*n* = 1)	Retail (*n* = 1)
		Korean (*n* = 1)	
		European (Other) (*n* = 1)	

^a^ Age data missing for one participant.

**Table 3 nutrients-16-00780-t003:** Co-design workshop objectives and activities.

Workshop Session	Objectives	Activities	RQ(s) ^a^
Session 1	Welcome Questions, queries, and concerns (PIS ^b^ and CF ^c^) Introductions—getting to know each other Exploration of the meaning of health and wellbeing Common barriers to achieving or maintaining healthy behaviours Thinking about health in relation to social media	Participants asked to introduce themselves and pick one ice breaker (e.g., what is the weirdest thing in your fridge right now? What is your go-to self-care activity? What food represents you and why?). Participants asked to reflect on what health and wellbeing means to them and invited to write their thoughts on the Zoom whiteboard. Participants asked to reflect on their barriers to achieving or maintaining healthy behaviours and invited to share with the group via whiteboard. Participants shown different nutrition-related social media content and asked to share their initial thoughts. After the session, participants are sent a brief screentime survey to determine their preferred apps of use, time spent on popular apps, what social media is predominately used for, and which social apps they think the intervention should use.	1, 3
Session 2	Deep dive into barriers and how social media can help Discussion around social media apps and use Perceptions of social media as a platform for behaviour change—do you think it would work?	A fictious story of two young women, both experiencing health barriers identified in the previous session, is shared with the group. Participants were then placed into two breakout rooms and asked to brainstorm what goals they would set for these women. Screentime survey results were shared with the group and discussed, as well as conversation prompts such as “where do you get your information about health and nutrition?” and “have you used social media to change a behaviour before? If so, what features did you use for that?”. Participants were placed into the same breakout groups for a final brainstorm—how could we help the young women achieve the goals you have set for them using social media?	2, 3
Session 3	Looking at influencers Quick content poll Understanding how young adults communicate on social media Developing the Daily Health Coach Healthy Navigation Toolbox Thank you!	Participants given a list of popular health influencers on Instagram and TikTok. They are asked to look at a couple and jot down their initial thoughts about their content. Participants were then invited to share their thoughts with the group. Prompt reflection questions included “would this person influence or motivate me?”, “does their content make me feel better?”, and “have they created a safe space or sense of community?”. Participants asked to respond to live Zoom poll, asking about content preferences: photo vs. video content, how much information they are willing to read on social media, do they think it is more important for information to be funny or entertaining, and what form of social media is the beast way for of delivering health content. Participants asked to scan the QR code and join a Mentimeter activity. This was used to brainstorm the tone of social media as a group. Participants were asked to look at their keyboards and send through their mostly commonly used emojis, as well as common phrases used when communicating or posting. Participants placed into breakout rooms as asked to have a go at drafting what they believe the outline of the Daily Health Coach should be. They were asked to discuss the following: the name of the intervention (do you like it or should we change it?), which apps should be used, video or photo content, the tone of the intervention (funny, informal, formal, or serious), how often they would want to be contacted by the research team via DM, what type of content should be posted, and how we can ensure the intervention is inclusive of all body shapes, sizes, cultures and genders.	1, 2, 3, 4

^a^ Research Question(s); ^b^ Participant information sheet; ^c^ Consent form.

**Table 4 nutrients-16-00780-t004:** Weekly content themes.

Monday	Myth-Busting	Debunking common nutrition myths such as is a calorie a calorie, the ketogenic diet, fat-burning foods, and supplementation.
Tuesday	Simple Swaps	Simple and realistic swaps for each meal, baking and cooking to boost micronutrient and fibre content of dishes and snacks.
Wednesday	Things You Should Know	Nutrition education including ultra-processed foods, nutrient-rich fats and carbohydrates, the gut microbiome, and the importance of dietary variety.
Thursday	The Other Important Stuff	Conversations around exercise (finding what works for you), sleep, stress, mental wellbeing, self-care, intuitive eating, and holism.
Friday	Real Talk	Important topics of conversation suggested by co-designers including beauty standards and norms, restrictive or clean eating, food relationships (moralisation of food), and using the scales.
Saturday	Healthy Navigation	Suggestions to tailor social media use to increase digital health literacy, improve body image, and avoid damaging or triggering content.
Sunday	Recipe Sharing	Each content creator has shared a recipe that is special to them, including the price per recipe, per shop, and per meal. Recipes are affordable and balanced.

## Data Availability

The data presented in this study are available on request from the corresponding author.

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
