# Peer review of "Empowering Young Women: A Qualitative Co-Design Study of a Social Media Health Promotion Programme"

_nutrients, 2024, doi:10.3390/nu16060780_

Round 1

Reviewer 1 Report

Comments and Suggestions for Authors

I commend the authors for conducting and summarizing such a comprehensive and seemingly well-executed research project.

I just have a few comments/suggestions:

It is mentioned that potential drawbacks of social media, such as promoting harmful body ideals, were taken into consideration, and addressed by content creators in a natural manner. However, were individuals with larger body types included in the evaluation of the content? This is important because perceptions of body image and fat shaming can vary depending on one's body type (i.e., weight status).

In the Discussion section, under subheading 4.3, there is a lengthy sentence that needs grammatical revision and appears to make an exaggerated claim about its novelty within its narrow context.

Moving on to subheading 4.4, the authors acknowledge that the individuals involved in the co-design/production were not representative of the cultural or socio-demographic diversity of young women in Aotearoa, New Zealand. This raises questions about what cultural differences might have emerged if co-designers/producers with Māori or Pasifika backgrounds had been included. Given that this journal caters to an international audience, it would be helpful to provide context.

Author Response

Thank you to reviewer 1 for their time and suggestions. Authors agree with the comments made and have addressed them within the manuscript and highlighted changes in yellow. Responses to comments have been specified in the attachment.

Reviewer 2 Report

Comments and Suggestions for Authors

The article aims to co-design a health promotion program for young women in Aotearoa using social media. The research method is a co-participatory design, using co-design guidelines published by NPC and the Young and Well Research Co-Operative Centre were used to develop the participatory framework for the development oft he Daily Health Coach intervention.

The results are very well presented and the conclusions are relevant for the advancement of the knowledge.

I would suggest that in the first part of the article the main objective of the paper to be more clear defined. Also section 4.1. ( Principal Findings) can and must be expanded by stressing the results of the analysis. 

The same is the case with the conclusions section which are very very short. This section have to be expanded for he benefit the the entire analysis

Author Response

Thank you to reviewer 2 for their time and suggestions. Authors agree with the comments made and have addressed them within the manuscript and highlighted changes in yellow. Responses to comments have been specified in the attachment.
